# Isolation and Characterization of Two Pseudorabies Virus and Evaluation of Their Effects on Host Natural Immune Responses and Pathogenicity

**DOI:** 10.3390/v14040712

**Published:** 2022-03-29

**Authors:** Qiongqiong Zhou, Longfeng Zhang, Hongyang Liu, Guangqiang Ye, Li Huang, Changjiang Weng

**Affiliations:** 1State Key Laboratory of Veterinary Biotechnology, Division of Fundamental Immunology, Harbin Veterinary Research Institute, Chinese Academy of Agricultural Sciences, Harbin 150069, China; qqhenanly@163.com (Q.Z.); yicunsangzhe@126.com (L.Z.); lhyyyds1234@163.com (H.L.); ygqyyds123@163.com (G.Y.); 2College of Animal Science, Yangtze University, Jingzhou 434000, China; 3Key Laboratory of Veterinary Immunology of Heilongjiang Provincial, Harbin 150069, China

**Keywords:** pseudorabies virus, virus isolation, pathogenicity, mortality, inflammatory response

## Abstract

Pseudorabies, caused by the pseudorabies virus (PRV), is an acute fatal disease, which can infect rodents, mammals, and other livestock and wild animals across species. Recently, the emergence of PRV virulent isolates indicates a high risk of a variant PRV epidemic and the need for continuous surveillance. In this study, PRV-GD and PRV-JM, two fatal PRV variants, were isolated and their pathogenicity as well as their effects on host natural immune responses were assessed. PRV-GD and PRV-JM were genetically closest to PRV variants currently circulating in Heilongjiang (HLJ8) and Jiangxi (JX/CH/2016), which belong to genotype 2.2. Consistently, antisera from sows immunized with PRV-Ea classical vaccination showed much lower neutralization ability to PRV-GD and PRV-JM. However, the antisera from the pigs infected with PRV-JM had an extremely higher neutralization ability to PRV-TJ (as a positive control), PRV-GD and PRV-JM. In vivo, PRV-GD and PRV-JM infections caused 100% death in mice and piglets and induced extensive tissue damage, cell death, and inflammatory cytokine release. Our analysis of the emergence of PRV variants indicate that pigs immunized with the classical PRV vaccine are incapable of providing sufficient protection against these PRV isolates, and there is a risk of continuous evolution and virulence enhancement. Efforts are still needed to conduct epidemiological monitoring for the PRV and to develop novel vaccines against this emerging and reemerging infectious disease.

## 1. Introduction

Pseudorabies (PR, Aujeszky’s disease), caused by the pseudorabies virus (Aujeszky’s disease virus or Suid alphaherpesvirus 1) (PRV), brings substantial economic losses to swine factories in some countries. The PRV belongs to the family Herpesviridae, subfamily Alphaherpesvirinae (https://talk.ictvonline.org/ictv-reports/ictv_online_report/dsdna-viruses/w/herpesviridae/1609/subfamily-alphaherpesvirinae, accessed on 10 Mar 2022). The PRV genome is a double-stranded DNA of about 143 kb in length, and the G+C content is more than 70% [1]. The genome of the PRV encompasses a unique long (UL) segment and a unique short (US) region flanked by the internal and terminal repeat sequences (IRS and TRS, respectively), encoding more than 70 proteins [2].

Since the early 1980s, PR disease had spread nearly globally. In 1947, PRV infection was reported firstly in eastern China. Because of the lack of accurate detection technology and poor biosafety measures, PR was widely prevalent in most of China. Owing to the wide usage of the traditional vaccine strain Bartha-K61 (a live attenuated vaccine based on the PRV Bartha-K61 strain), PR diseases were controlled in China [3]. However, in 2011, the outbreaks of PR were reported and rapidly spread in China. In some farms, the PRV seropositive occurred in a very short time. Research confirmed that the current PR outbreaks were caused by PRV variants and the Bartha-K61 vaccine could not provide adequate protection against these variant strains [3,4]. Subsequently, several PRV isolates were isolated in China, which were in the same evolutionary branch as the JS strain (KP257591), and in two independent branches with the European and American strains (Kapla, Becker) [5]. The variation regions of PRV isolates are mainly concentrated in IRS, TRS and SSR sequences in the promoter region, which can affect the gene expression and pathogenicity of PRV [6]. In addition, it was reported that the pathogenicity of PRV isolates to mice and pigs was significantly higher. For instance, the LD_50_ of SC strain was 10-fold and 8.5-fold higher than that of PRV-TJ (KJ789182) and PRV-HeN1 strains (KP098534.1), respectively [4]. In 2016, Yang et al. illustrated that PRV variant isolate HN1201 (KP722022) could cause extensive tissue injury and 100% death of piglets, but the Fa isolate (KM189913) only induced weak respiratory symptoms [7]. To better control and prevent PR in China, the Chinese government has issued a series of policies with the intent to eradicate PR in pig breeding farms by the end of 2020. The rational use of vaccines (including inactivated vaccines, live attenuated vaccines, live virus-vectored vaccines) and other novel viral inhibitors are the main strategies.

Without specific host tropism, the PRV infects a wide variety of animal species, including ruminants, carnivores, rodents, and lagomorphs. The genus sus scrofa are the only natural hosts for the PRV, which cause severe clinical symptoms [8]. The PRV can infect pigs at different ages, causing 100% death of piglets, abortion of sows, and respiratory diseases in adult pigs [9]. Aside from pigs, cattle, sheep, cats, dogs, raccoons, minks, and skunks, can all be infected by the PRV, causing “mad itch”, neurological symptoms, or death [10,11]. Moreover, it was demonstrated that the PRV infected host cells via both human and swine nectin-1, and that PRV glycoprotein D exhibited similar binding affinities for nectin-1 of two species [1]. Symptomatically, human PRV infection cases in China indicated that PRV infection could induce prominent central nervous system disorders and encephalitis [12,13], which posed a significant threat to public health in China [12,13,14,15]. Therefore, it is important to improve the phylogenetic analysis and virulence monitoring of PRV variant isolates.

The immune system is the most significant line of host defense against virus infection. Viral infection is defended by hosts using multiple strategies, including innate immune responses and adaptive immunity. The inflammatory response is a complex mechanism, consisting of immune cells and inflammatory cytokines (e.g., IL-1β, IL-6, TNF-α, MCP-1, IP-10, MIP-1, etc.), which remove invading viruses and promote repair at the sites of damage [16,17]. The type I Interferon (IFN-I) response is the most prominent antiviral response, which plays a central role in innate immunity against viral infection. IFN-I-targeting cells maintain a potent antiviral state by inducing the synthesis of hundreds of antiviral proteins encoded by IFN-stimulated genes (ISGs). Increasing evidence indicates that the PRV has evolved multiple strategies to inhibit type I IFN signaling and establish persistent infection [18,19].

The PRV is an emerging and reemerging infectious disease consistently threatening the pig industry worldwide. As the largest pork producer and consumer, China has experienced two pseudorabies outbreaks [20]. Since 2011, more and more PRV variant isolates, whose gene types belong to Clade 2 (variant PRV), were identified in China [13]. It was reported that the sero-prevalence rate of PR was 34.2% in China from 2016 to 2018. Furthermore, the sero-prevalence in northern China, eastern China, and central and southern China are higher than those in northeastern China, northwestern China, and southwestern China [20]. Notably, increasing number of PRV-infected human cases were reported [13,14,15]. Given the current global epidemic of PRV variant strains, herein, two fatal PRV (PRV-GD and PRV-JM) were isolated, and the etiological as well as genetic characteristics of these PRV isolates were investigated. Moreover, the pathogenicity to mice and pigs and the natural immune responses induced by these PRV isolates were also explored in vitro and in vivo.

## 2. Materials and Methods

### 2.1. Isolation of PRV-GD, PRV-JM and Genome Sequencing

PRV-GD and PRV-JM were isolated from the aborted piglet samples of PRV-positive pig farms at Foshan and Jinmen, Guangdong Province, respectively. The PRV-GD and PRV-JM isolates, purified by plaque, were inoculated onto PK-15 cells. To confirm the occurrence of virus multiplication, the one-step growth curve of the PRV-TJ, PRV-GD, and PRV-JM isolates in PK-15 cells and the *gB* gene expression were assessed.

The genomic DNA of PRV-GD and PRV-JM isolates were extracted from the infected PK-15 cells. DNA quality, integrity, and concentration were assessed, and sequencing was performed by Harbin Biotech Gene Company (Harbin, China). Then, the nucleotide sequences of two virus were compared with known PRV strains retrieved from the GenBank database. Phylogenetic tree analysis was constructed using MEGA 7.0 software. 

### 2.2. Virus Neutralization Assay

The positive antisera, collected from different pigs immunized with classic PRV-Ea live attenuated vaccines, were used for the neutralization assay as previously described [21]. Briefly, the antisera were heat-inactivated for 30 min at 56 °C and serially diluted from 2^0^–2^−7^ in 2-fold. The neutralization tests were conducted by adding 50 μL (containing 100 TCID_50_) of virus suspension into 50 μL of the diluted antisera in 96-well plate in quadruplets, and then incubated at 37 °C for 40 min. Subsequently, the 100 μL mixtures were added into 96-well plate and incubated with PK-15 cells at 37 °C incubator to observe the cytopathic effect (CPE). The neutralizing titer was expressed as the highest dilution that reduced the CPE by 50% as compared to the control. The neutralizing titer was calculated based on the Reed–Muench method [16]. 

### 2.3. Immune Responses in Mouse Peritoneal Macrophages Induced by PRV Strains

The PRV can infect a wide variety of mammals, including pigs, wild boars, rodents, bears, ruminants, and carnivores. In general, PRV-variant infections are fatal to rodents [8]. To evaluate the pathogenicity and immune responses of PRV isolates more comprehensively, the mice and piglets were used to perform the experimental infection study.

Firstly, we detected the innate immune responses in mouse peritoneal macrophages induced by PRV-GD, JM, and TJ strains. Primary peritoneal macrophages were isolated from C57BL/6J mice and infected with PRV-GD, JM and TJ strains for 0, 3, 6, 9, 12, and 24 h. The total mRNA and the cell culture supernatant were harvested. The transcription levels and the protein levels of multiple pro-inflammatory cytokines, such as interleukin-1β (IL-1β), IL-6, tumor necrosis factor-α (TNF-α), and interferon-β (IFN-β) were detected.

### 2.4. Experimental Mice Infection Study

Then, a total of 40 six-week-old SPF C57BL6/J mice were purchased from Liaoning Changsheng Biotechnology Co., Ltd. (Liaoning, China). All mice were generated and housed in specific pathogen-free (SPF) barrier facilities at the Harbin Veterinary Research Institute (HVRI) of the Chinese Academy of Agricultural Sciences (CAAS) (Harbin, China) (the ethical approval number: 210608-01). All mice were randomly divided into eight groups (*n* = 5, respectively). Three groups were challenged with PRV-GD (*n* = 5) and another three groups were challenged with PRV-JM (*n* = 5) isolates at different doses by intraperitoneal injection (i.p.). The remaining two groups received PBS (50 mM, pH 7.4) injections. Mortality in each group was recorded and LD_50_ was calculated based on the Reed–Muench method [16]. The liver, spleen, lung, kidney, and brain samples were collected and used for H&E staining.

Furthermore, to detect the effects of PRV-GD and PRV-JM isolates on the natural immune response of mice, nine six-week-old SPF C57BL6/J mice were randomly divided into 3 groups (*n* = 3, respectively). Three groups of mice were challenged with 20,000 PFU of PRV-GD, PRV-JM isolates, or PBS via tail vein injection. Two days post-infection, all mice were executed by a cervical vertebrae luxation after CO_2_ inhalation anesthesia. The serum, liver, spleen, lung, kidney, and brain samples were collected and detected for IL-1β, IL-6, TNF-α, IFN-β by ELISA or RT-qPCR, respectively.

### 2.5. Pathogenicity in Piglets

Eight crossbred healthy SPF piglets (Landrace × large white, 2-month-old, male, 9~11 kg) were purchased and housed in SPF barrier facilities at the Harbin Veterinary Research Institute (HVRI) of the Chinese Academy of Agricultural Sciences (CAAS) (Harbin, China) (the ethical approval number: 211026-02). All piglets were maintained at an ambient temperature of 20–25 °C in an environmentally controlled room by air conditioning and illumination (12 h light and dark cycles). Each cage was equipped with a feeder and water nipple to allow free access to food and drinking water. 

Eight two-month-old piglets were randomly divided into three groups. Pigs in groups 1 and 2 (*n* = 3, respectively) were inoculated intranasally with PRV-GD or PRV-JM at 10^6^ TCID_50_. Pigs in group 3 (*n* = 2) received PBS (50 mM, pH 7.4) as the control. Rectal temperature and clinical signs (including diet, water intake, mental status, and neurological symptoms) were recorded daily. Six days post-infection, all PRV-infected pigs died and were necropsied. The PBS-changed pigs were euthanized (electric shock anesthesia). The serum, liver, spleen, lung, kidney, tonsil, and brain samples were collected and detected for IL-1β, IL-6, TNF-α, IFN-β, IFN-α by ELISA or RT-qPCR, respectively. The samples also were used for H&E and virus detection using *gB*-specific TaqMan qPCR.

### 2.6. Histological Analysis

The tissues (liver, spleen, lung, kidney, tonsil, and brains) of mice and piglets (the tonsils were taken from pigs only) infected with PRV isolates were fixed in 10% formalin neutral buffer solution overnight. Histological analysis of tissue damage was assessed by standard hematoxylin and eosin (H&E) staining. Tunnel solution (Beyotime) was used to detect the damaged nucleus. The results were analyzed by light microscopy. Representative views of the tissue sections are shown.

### 2.7. Statistical Analysis

Data were analyzed as mean ± SEM of at least three independent replicates. Differences among groups were performed by one-way ANOVA using GraphPad Prism software. *p* values of < 0.05 were considered to be statistically significant for each test. The significance level for all analyses was set as * *p* < 0.05, ** *p* < 0.01 and *** *p* < 0.001.

## 3. Results

### 3.1. Identification of PRV-GD and PRV-JM Isolates

Cytopathic effects (CPE) induced by PRV-GD and PRV-JM, including cell fusion, syncytium, detachment, and numerous rounded and floated cells, were observed at 24 h post-infection (hpi). The CPE in PRV-TJ-infected PK-15 cells were considered as the positive and no CPE were observed in the control (Figure 1A). As shown in Figure 1B, PRV glycoprotein B (*gB*) and glycoprotein E (*gE*) genes were detected by polymerase chain reaction (PCR) amplification to confirm the existence of PRV in cell culture. The results of Figure 1C showed that there was no significant difference in the growth of the three PRV strains on PK-15 cells.

### 3.2. Genome Sequencing and Phylogenetic Analysis

The genome of PRV-GD and PRV-JM isolates were extracted and sequenced by second generation sequencing technology. The complete genomes of PRV-GD (GeneBank accession numbers OK338076) and PRV-JM (GeneBank accession numbers OK338077) isolates are 144.05 kb and 142.47 kb, respectively. To further understand the evolutionary relationship between the two PRV isolates and other PRV variants, a phylogenetic tree was constructed based on their genomes (Figure 1D). The results indicated that PRV-GD and PRV-JM isolates, belonging to genotype 2.2, were novel PRV variants. These results demonstrated that the population size of PRV clade 2.2 was increasing, indicating that PRV variants may be still circulating in swine herds and result in a risk in relation to interspecies transmission.

### 3.3. Proliferation Characteristics of PRV-GD and PRV-JM in PAMs, THP-1 and Mouse Peritoneal Macrophages

One-step growth curves were performed to evaluate the proliferation characteristics of PRV-GD and PRV-JM in PAMs, THP-1, and mouse peritoneal macrophages. The PRV-TJ-infected cells were considered as the positive control. Firstly, virus-induced CPEs in three cells were compared. PAMs, THP-1, and mouse peritoneal macrophages were challenged with PRV-TJ, PRV-GD, and PRV-JM. PAMs, THP-1, and mouse peritoneal macrophages showed the similar CPEs, characterized by shrinkage, fragmentation, and cell death, but mouse peritoneal macrophages showed fibrosis and death. Compared with PRV-GD and PRV-TJ, PRV-JM induced stronger CPEs in the test cell lines (Figure 2A). In one-step growth analysis, PRV-GD and PRV-JM displayed similar growth curves as the PRV-TJ strain in both PK-15 cells and PAMs (Figure 1C and Figure 2B). However, in THP-1 and mouse peritoneal macrophages, PRV-GD and PRV-JM isolates showed a replication superiority compared with the PRV-TJ strain (Figure 2C,D). These observations together supported our proposal that PRV-GD and PRV-JM isolates were the novel PRV variants and behaved differently from other classical PRV variants.

### 3.4. Cross-Neutralization Assays

Next, antibody neutralization assay was performed to determine the immunogenicity of PRV-GD and PRV-JM isolates. As shown in Appendix A, the antisera from sows immunized with classical Ea vaccination showed strong seropositive against PRV *gB* protein. Subsequently, the antisera exhibited higher neutralization activity to PRV-TJ but weaker neutralization ability to PRV-GD and PRV-JM (Figure 3A). However, antisera from pigs infected with PRV-JM showed extremely high neutralization activity to PRV-TJ, PRV-GD and PRV-JM (Figure 3B). Collectively, these results suggest that the traditional vaccine prepared with classical PRV strain may not provide effective protection against the challenge of PRV variants, suggesting there is a potential risk of increasing virulence of PRV variants from the perspective of immunogenicity.

### 3.5. Immune Responses Induced by PRV Strains

The mRNA levels of multiple pro-inflammatory cytokines, such as interleukin-1β (IL-1β), IL-6, tumor necrosis factor-α (TNF-α), and interferon-β (IFN-β), were up-regulated during the three PRV variants infection (Figure 4A). Consistently, ELISA results illustrated that the protein levels of these pro-inflammatory cytokines and IFN-β were also increased (Figure 4B). Additionally, after infection with three PRV strains, the transcriptional levels of IL-1β, IL-6, TNF-α, and IFN-β were highest at 3 h, 6 h, 9 h, and 12 h, respectively, and then gradually decreased. The protein levels of IL-1β, IL-6, and IFN-β were highest after infection of 12 h, but TNF-α was maintained in the higher levels after 24 h (Figure 4). It was noteworthy that PRV-GD and PRV-JM induced higher levels of the pro-inflammatory cytokines and IFN-β than PRV-TJ strain, eventually leading to stronger immune responses.

### 3.6. Pathogenicity of PRV-GD and PRV-JM Isolates in Mice

To gain the pathological insight into the overall changes induced by PRV-GD and PRV-JM, pathological examinations in mice were firstly performed. The SPF C57BL/6J mice were infected with two isolates by intraperitoneal injection. As shown in Figure 5A,B, both PRV-GD and PRV-JM infection could induce itch, eventually leading mice to death with a comparable LD_50_ (50% lethal dose): 57.5 and 66.1 TCID_50_ (50% tissue culture infective dose), respectively (Figure 5A,B). Subsequently, dissection of the mice immediately after death was performed to analyze the pathological characteristics by H&E staining. The results showed that PRV-GD and PRV-JM isolates could significantly cause lung congestion, thickening of alveolar septa, lymphocyte infiltration in the spleen, disintegration of hepatocytes, cellular necrosis in the liver and kidney, and microglia proliferation in the brain (Figure 5C).

In addition, three groups of SPF mice were infected with PRV-GD and PRV-JM isolates at 20,000 PFU/mL to examine the innate immune responses. The results were summarized in Appendix A and Figure 5D. Among the examined tissues (liver, spleen, lung, and brain), both PRV-GD and PRV-JM could enhance the expression of pro-inflammatory cytokines (IL-1β, IL-6, TNF-α) and IFN-β after 2 days post-infection (dpi). The lung possessed the highest PRV genomic copy number among the examined tissues (Appendix A). Of note, the two isolates induced similar patterns of changes of the pro-inflammatory cytokines and IFN-β in serum by ELISA (Figure 5D). Overall, our results indicate that PRV-GD and PRV-JM are the virulent isolates with high pathogenicity and can induce robust inflammatory responses in mice. 

### 3.7. Pathogenicity of PRV-GD and PRV-JM Isolates in Pigs

To further gain insight into the pathogenicity and inflammatory responses of PRV-GD and PRV-JM isolates in pigs, two-month-old piglets were treated with PRV-GD or PRV-JM at 10^6^ TCID_50_, and the innate immune responses and cell death were monitored. Indeed, piglets treated with PRV-GD or PRV-JM initially (48 h) exhibited labored breathing, hydrostomia and reduced feed intake, accompanied by a high fever (>40.5 °C) (Figure 6A). Subsequently, piglets became severely dyspnea, lost weight (Figure 6B), and exhibited neurological symptoms, including convulsion, ataxia, and paddling, and all piglets died after 6 dpi (Figure 6C). The pathological change results showed that PRV-GD and PRV-JM caused hyperemia in the brain, tonsils and kidneys, and necrosis in the lungs and liver (Figure 6D). 

Additionally, qPCR and ELISA were executed to detect the mRNA and protein levels of inflammatory cytokines in piglets infected with PRV-GD and PRV-JM. As structured in Appendix A, the transcription level of pro-inflammatory cytokines (IL-1β, IL-6) and IFN-β were enhanced in PRV-GD and PRV-JM-infected tissues; nevertheless, a weaker change happened in TNF-α. The highest viral load was in the tonsils, while the spleen showed the lowest viral load (Appendix A), suggesting that the tonsils may be the main organ for PRV proliferation in pigs. The ELISA results demonstrated that PRV-infection induced a high level of IFN-α and IFN-β, but a weak release of IL-1β after PRV infection for 2 d and 5 d in pigs (Figure 6E–G). It was noted that there was no IL-6 in serum after PRV infection for 2 d and 5 d (data not shown). Furthermore, the PRV genomic copy number in the blood was significantly enhanced after 2 dpi and then gradually increased (Figure 6H). 

Moreover, pathological examinations demonstrated that PRV-GD and PRV-JM infection resulted in glial cell proliferation, neuron degeneration, inflammatory cell infiltration in brain, cell necrosis, congestion in the tonsils, kidneys and spleen, as well as extensive serous exudation, inflammatory cell infiltration, and alveolar epithelial cell abscission in the lung (Figure 7A). Tunnel labeling results suggested that PRV-GD and PRV-JM infection induced cell death (red signal) in the tonsils and brain, and no signal was observed in the PBS-treated piglets (Figure 7B,C).

## 4. Discussion

Since the early 1980s, PR had spread almost globally, mainly as sequelae of the international movement of animals and animal products. Due to strict animal disease control measures and eradication programs, including the extensive usage of the traditional vaccine strain Bartha-K61, PR has virtually disappeared from domestic pigs in several countries, e.g., Austria, Germany, Switzerland, Great Britain, Canada, New Zealand, and the United States [22]. However, PR is still endemic in areas with dense pig populations, i.e., some regions in eastern and south eastern Europe, Latin America, Africa, and Asia [22,23]. In addition, wild boar was a potential and persistent reservoir for PRV, since PRV-infected wild boar represent an increasingly obvious threat for the reemergence of PRV into free regions [8,23]. As the largest swine breeding country, highly pathogenic PRV variant epidemics have caused huge economic losses and great difficulties in the prevention and control of PR in China since 2011 [24,25]. In our study, we successfully isolated and identified two new PRV variant strains from the PRV seropositive pig farms (Figure 1). Phylogenetic trees were constructed to reveal the origin and genetic relationships between the two PRV isolates and other strains (Figure 1D). Animal experiments indicated that PRV-GD and PRV-JM isolates could lead to a high mortality in mice and piglets, as well as adult pigs (data not shown). This study provides new information about the prevalence of the PRV variants currently circulating in China.

Symptomatically, PRV infection results in prominent central nervous system disorders and acute encephalitis in both humans and animals [26]. It was reported that PRV-infected pigs could spread PRV to healthy pigs, people, sheep, raccoons, and vice versa. However, the horizontal spread within non-natural hosts might not exist [13]. Therefore, pigs are the only reservoir host of PRV and are recognized as the central link of PRV cross-species transmission. Recombination of PRV strains in pigs may result in changes of antigenicity, virulence, and thus immune failure, which could be the source of continuing epidemics. Therefore, strengthening the monitoring of the prevalence of PRV variants in pigs has great significance in preventing and controlling the spread of PRV variants among other species.

Previous studies showed that PRV strains can be divided into two main clades with frequent interclade and intraclade recombination. Clade 2.2 (PRV variant) is currently the most prevalent genotype worldwide that is most frequently involved in interspecies transmission events (including humans) [13]. According to recent studies, novel PRV variants showed enhanced pathogenicity. Evidence revealed that only 40% of human genes was found to utilize translation initiation sites (TISs) [27]. In comparison, PRV could highly efficiently utilize TISs and integrate the genes of hosts or other viruses, providing more additional possibilities for PRV genetic variation. Indeed, PRV variants delineated a more complex transcriptome and identified an unexpectedly large number of potential novel genes [28]. Additionally, mounting research suggests that overlapping transcription may be the novel strategies of the PRV to regulate its gene expression, escape host innate antiviral immunity, and fulfill genetic evolution at different infection stages [27,29]. In this study, although in the same genetic branch, PRV-GD and PRV-JM isolates had stronger virulence and proliferation rate than that of PRV-TJ on THP-1 cells and mouse peritoneal macrophages (Figure 2). Notably, pigs immunized with classical Ea vaccine were incapable of providing sufficient protection against the two novel PRV-GD and PRV-JM isolates, whereas the antisera from pigs infected with PRV-JM had a high cross-neutralization activity to PRV-TJ, PRV-GD, and PRV-JM (Figure 3). These results suggest that PRV-GD and PRV-JM are virulent isolates and further research is needed for the enhanced virulence. In general, there is an urgent need to strengthen epidemiologic surveillance and develop new effective vaccines to control PR by targeting variant isolates.

In 1970, Jentzsch reported that PR could not occur in humans [30]. Recently increasing human viral infection cases involved in PRV have been reported in China, indicating that PRV can spread from pigs to humans and result in pruritus, throat pain, disturbance of consciousness, or respiratory failure [13]. It makes sense that the virus can spread between humans and swine, because there is a 96% homology of nectin-1 receptor between humans and swine, which mediate several viruses (PRV, HSV-1 and BHV-1) to enter cells [31,32]. In our study, compared with PRV-TJ strain, PRV-GD and PRV-JM isolates had a similar multiplication rate in the PK-15 cells and PAMs but showed enhanced proliferation in THP-1 cells and mouse peritoneal macrophages (Figure 2), suggesting PRV-GD and PRV-JM may have the stronger appetency to these cells. Furthermore, pigs have considerable impacts on human health because of the high similarity of the anatomical structures and immune systems, as well as promising medical resources in xenotransplantation [33]. At present, there are no effective drugs to prevent the progression of the disease caused by PRV infection. Therefore, although human PRV infection cases were rare, it is not be ignored that PRV pose a significant threat to public health, especially in people in close contact with sick pigs and/or related pork products/contaminants. In addition, whether PRV variant strains have enhanced virulence to humans and whether PRV-infection induce a cross-protection against others herpes viruses are also unknown and need to be further studied. 

A higher level of pro-inflammatory cytokines (IL-1β, IL-6, TNF-α) were induced in serum from the PRV-infected mice, while only a small amount of IL-1β, and no IL-6 were detected in the serum from the PRV-challenged pigs (Figure 5D and Figure 6G). In addition, PRV infection causes meningitis and conjunctivitis in humans, severe pruritus in mice, and severe respiratory symptoms in adult pigs. These investigations indicated that there were great differences to the hosts’ innate immune responses induced by PRV among different species, which may be due to the differences in the immune systems of rodents and pigs. Thus, for pigs or mice, it is worth exploring which is the best model animal to study the pathogenicity of PRV in humans. The detailed molecular mechanisms also need to be further explored.

In summary, two fatal PRV strains, PRV-GD and PRV-JM, were isolated and characterized as the novel variant isolates according to their etiological features and phylogenetic relationships, as well as the lethal rate to mice and pigs. Given the current global epidemic of PRV variant strains in pigs, our analysis of the PRV variants emergence illustrates the need for continuous monitoring and the development of vaccines against specific variants of PRV.

## Figures and Tables

**Figure 1 viruses-14-00712-f001:**
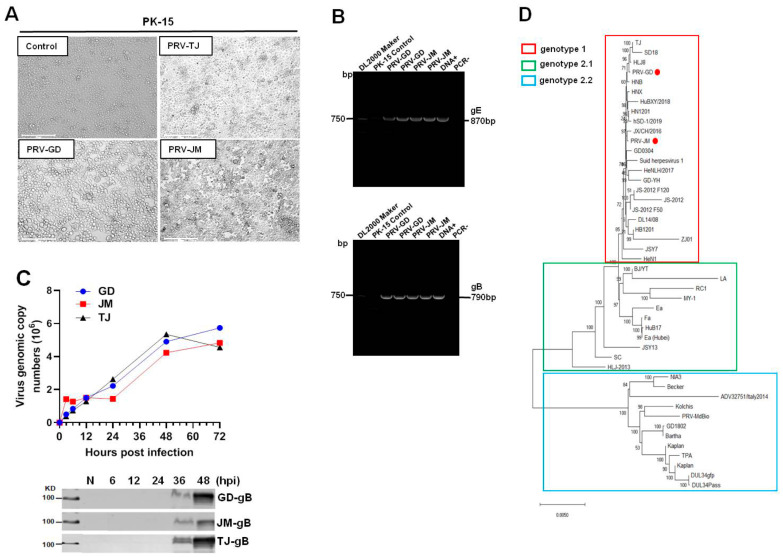
Isolation and identification of PRV-GD and PRV-JM isolates. (**A**), The cytopathic effects (CPEs) of PK-15 cells infected by PRV-GD and PRV-JM for 24 h. The CPEs of PK-15 cells infected with PRV-TJ strain were considered as the positive control. The arrowhead indicates the CPEs observed of PRV-GD and PRV-JM-inoculated PK-15 cells. The CPEs were characterized with rounded and floated cells. (**B**), PCR amplification of PRV *gE* (870 bp) and *gB* (790 bp) fragments from PRV-GD and PRV-JM-inoculated PK-15 cells. DNA+ was the positive control and PCR− was the negative control during PCR amplification. (**C**), One-step growth curves and *gB* protein assessment of 3 PRV strains on PK-15 cells at a multiplicity of infection of 0.1. (**D**), Phylogenetic tree based on genomic nucleotide sequence. PRV-GD and PRV-JM isolates in this work were indicated with a red dot. The phylogenetic tree was constructed by the adjacency method in MEGA 7 (http://www.megasoftware.net, accessed on 14 July 2021).

**Figure 2 viruses-14-00712-f002:**
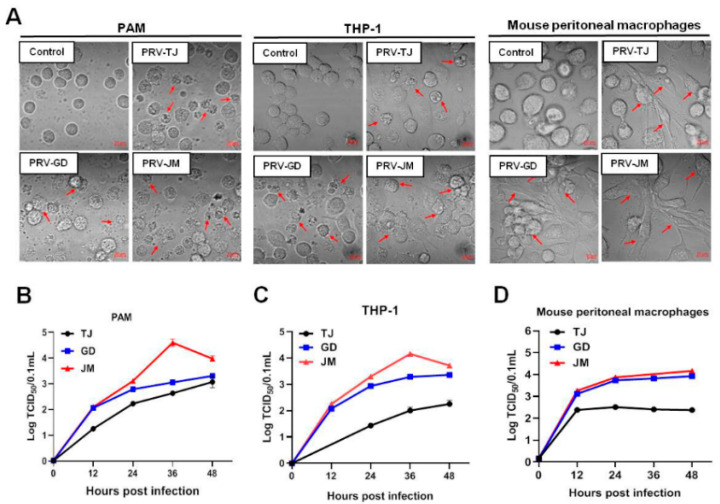
In vitro proliferation properties of PRV-TJ strain, PRV-GD and PRV-JM isolates. (**A**), CPEs in PAMs, THP-1 cells and mouse peritoneal macrophages, caused by PRV-TJ strain, PRV-GD and PRV-JM isolates, were characterized by the disintegrated cells (red arrows). Scale bars: 10 μm. (**B**–**D**), One-step growth curves of PRV-TJ, PRV-GD and PRV-JM on PAMs, THP-1 cells and mouse peritoneal macrophages at a multiplicity of infection of 0.1.

**Figure 3 viruses-14-00712-f003:**
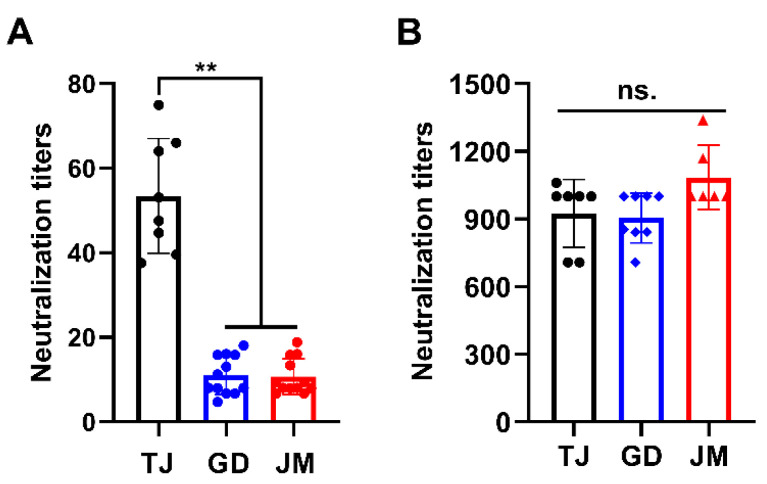
Antisera from pigs infected with PRV-JM isolate has broader spectrum of neutralizing ability. Neutralizing titers of antisera from classical PRV-Ea-vaccinated sows (**A**) and antisera from pigs immunized with PRV-JM isolate (**B**) against PRV-TJ, PRV-GD and PRV-JM. The significance of differences between the PRV-TJ, PRV-GD and PRV-JM were analyzed with *t* test. ** *p* < 0.01. ns., no significant difference.

**Figure 4 viruses-14-00712-f004:**
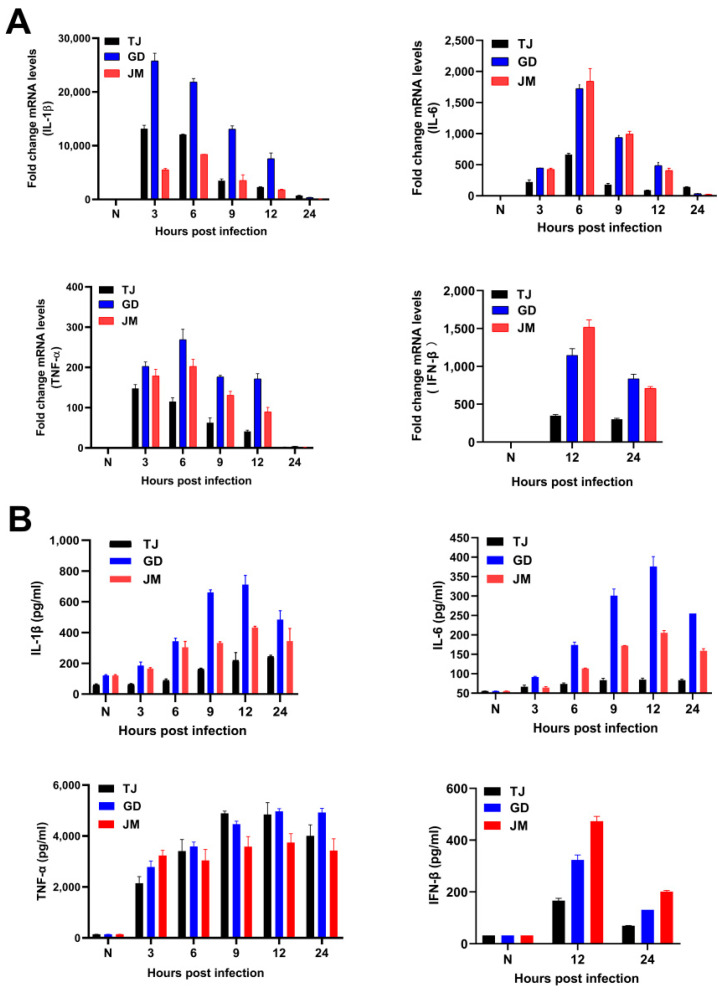
PRV-GD and PRV-JM isolates induced the higher immune responses than PRV-TJ strain. (**A**,**B**), Detection of the mRNA and protein levels of the IL-1β, IL-6, TNF-α, and IFN-β in mouse peritoneal macrophages induced by PRV-TJ, PRV-GD, and PRV-JM after infection for 3, 6, 9, 12 and 24 h, respectively.

**Figure 5 viruses-14-00712-f005:**
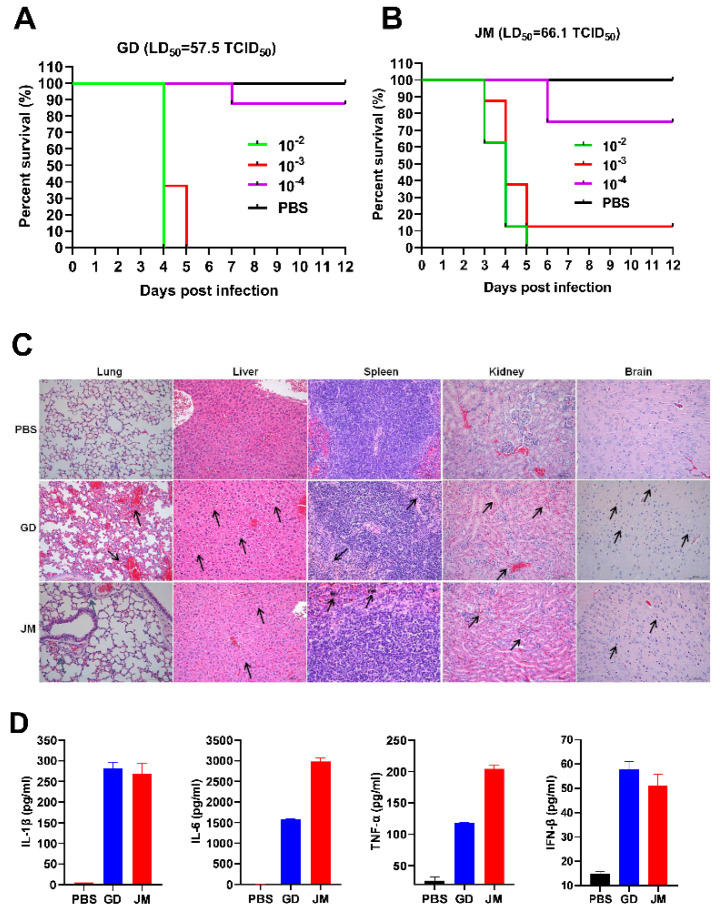
The pathogenicity and the immune responses of PRV-GD and PRV-JM in experimental mice. (**A**,**B**), LD_50_ of the PRV-GD (**A**) and PRV-JM (**B**) to six−week−old C57BL/6J mice. (**C**), Pathological lesions of mice (H&E staining, black arrow) that died following experimental infection with PRV-GD and PRV-JM isolates. (**D**), Detection of the protein levels of IL-1β, IL-6, TNF-α and IFN-β in serum challenged by PRV-GD and PRV-JM isolates.

**Figure 6 viruses-14-00712-f006:**
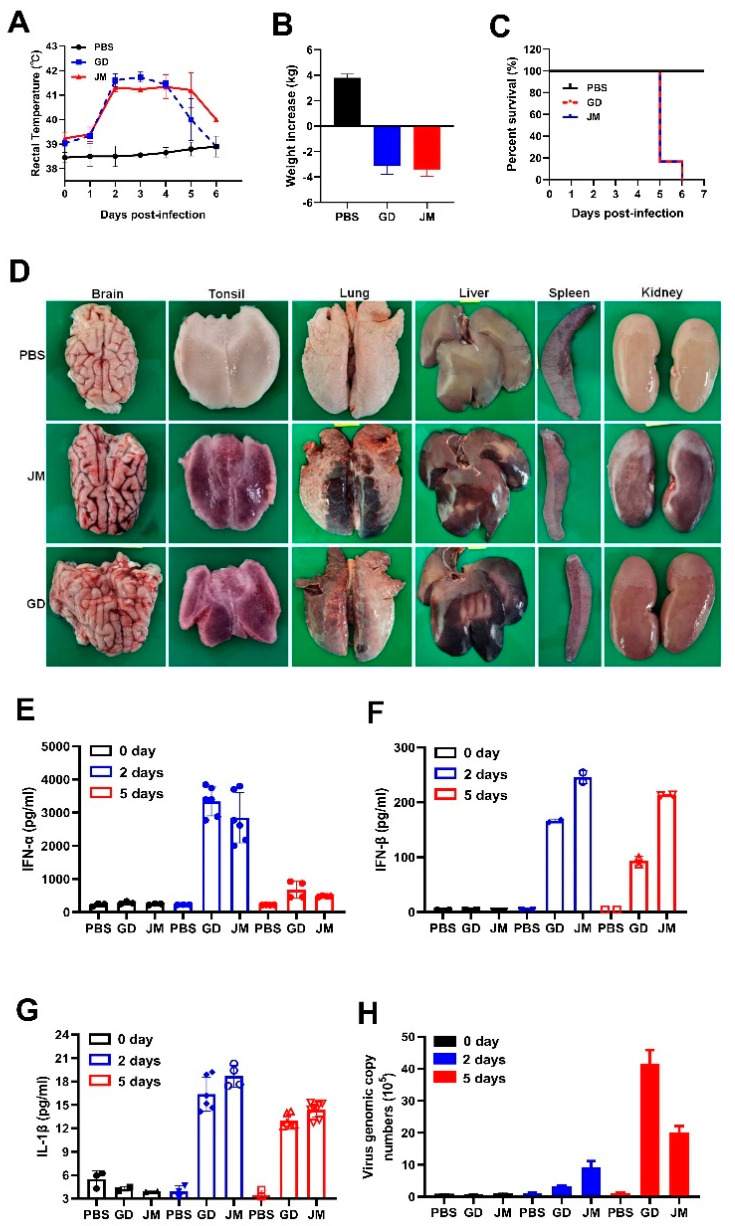
In vivo pathogenicity of PRV-GD and PRV-JM isolates in piglets. (**A**), Rectal temperature of the challenged piglets during the observation. (**B**), The changes of weight of the experimental piglets under the same raised conditions from pre-infection to death. (**C**), Survival rates of piglets infected with PRV-GD or PRV-JM isolates. (**D**), The tissue pathological changes of piglets. (**E**–**G**), Detection of the protein level of IFN-α, IFN-β and IL-1β in serum from piglets challenged by PRV-GD or PRV-JM. (**H**), Detection of the PRV genomic copy number in blood after infection for 0 days, 2 days and 5 days.

**Figure 7 viruses-14-00712-f007:**
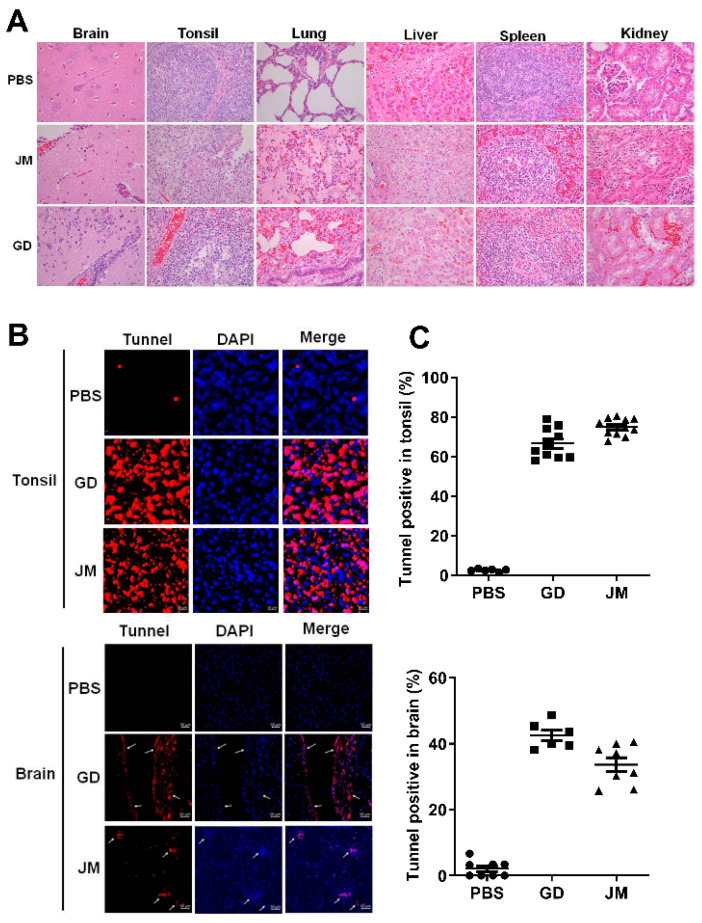
PRV-GD and PRV-JM infection induced tissue injury and cell death. (**A**), Pathological lesions of piglets (H&E staining) that died following infection with PRV-GD or PRV-JM isolate at 10^6^ TCID_50_. (**B**), Tunnel solution was used to label the dead cells in the tonsils and brains of piglets. (**C**), The percentage of tunnel-labeled cells was quantified.

## Data Availability

Data supporting the reported results are available in this article and in the Appendix A.

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
