# Peer review of "Isolation and Characterization of Two Pseudorabies Virus and Evaluation of Their Effects on Host Natural Immune Responses and Pathogenicity"

_viruses, 2022, doi:10.3390/v14040712_

Round 1

Reviewer 1 Report

A well-designed comprehensive study. Congratulations to the working team.

It would be more appropriate to clearly indicate the location of the new isolates in picture 1d.

Author Response

Review 1

Q1. A well-designed comprehensive study. Congratulations to the working team.

Authors’ Response:

We are grateful to you for the careful review and overall favo­rable impression on this manuscript.

Q2. It would be more appropriate to clearly indicate the location of the new isolates in picture 1d.

Authors’ Response:

Thanks for your comments. According to your suggestion, the location of the new isolates in picture 1d were indicated with red dot in our revised manuscript.

Reviewer 2 Report

The article entitled "Isolation and characterization of two pseudorabies virus and evaluation of their effects on host natural immune responses and pathogenicity aims to evaluate the pathogenicity and the effect on immunity of two fatal variants of the pseudorabies virus. Both aspects were evaluated on vaccinated piglets and mice. The results show that the common vaccines used for prophylaxis are not able to protect pigs in the two variants examined. The work underlines the importance of monitoring the evolution of the different PRV variants and therefore of the risk of a probable enhancement of virulence. Aujeszky's disease still represents a major problem in pig industries for several countries due to severe production losses, as well as the ascertained zoonotic potential. While the control and vaccination strategy has allowed the eradication of the disease in some countries, in many others, such as China, it still represents a challenge due to the emergence of variants for which the common vaccines used could not be more effective. The purpose of the work is believed to be original as well as relevant to current scenarios. However, the article is lacking in several parts (see particular comments): the introduction should be focused on the variants currently circulating, in the text several times mention is made of "classical strain" but the reader is not given a summary of the current state of the art in China ; it is suggested to also argue about the type of vaccines used (with reference to the strain, if alive or attenuated) as well as about the current control strategies that are implemented in China. The study design is not clear and the paragraphs relating to the materials and methods and results section (see specific comments) should be better presented as there is a risk of confusing the reader as they do not clarify the aims; the statistical analysis of the data is mentioned in the materials and methods section but it does not indicate which data have been analyzed, and furthermore they are not included in the results;

the general evaluation of the article is average, and it is advisable to make a major review before publication.

Line 19: it is suggested to contextualize (area-country) with regard to the variants currently circulating.

Line 22: Regarding the PRV-TJ variant, specify that it is a positive control, as shown in the results section, furthermore this variant is not described in the materials and methods section (See the following comments)

Line 22: Regarding the PRV-TJ variant, specify that it is a positive control, as shown in the results section, furthermore this variant is not described in the materials and methods section (See the following comments)

Lines 41-55: It is suggested to improve the description of the state of the art in China, to indicate the variants currently circulating in fact, often reference is made to "classical strains", but it is not clear which ones are being referred to. In addition, it is recommended to provide an overview of the prevalence and incidence data of Aujeszky's disease in China and the typology of current control measures.

Lines 57-58: the statement "Pigs are only natural host" is not properly correct, it would be more appropriate to refer to the "genus Sus Scrofa", which also includes wild boar, moreover it is not strictly correct to say that it causes "an acute and fatal” referring to the“ pigs ”since, the fatality and the severity of the symptoms increases with decreasing with age, while it is correct for the unnatural hosts. Therefore, it is advisable to rephrase the entire period (lines 56-61) in order to avoid generating confusion in the reader.

Line 63: the abbreviation (gD) is superfluous considering that it is not used in the manuscript.

Lines 68-69: the period denotes a conclusion, so it is advisable to omit it in the introduction and move it to the "discussion" and / or "conclusions" sections.

Line 71: This sentence is superfluous; it is advisable to rephrase it.

71-81: it is suggested to clarify, that the description refers “in general” to a viral immune response, furthermore please check the reference n. 16 as it refers to influenza viruses. With regard to the inflammatory response and the cytokines mentioned, it is suggested to provide pertinent information to the pseudorabies virus and in this regard it is advisable to read the review "https://www.mdpi.com/1999-4915/14/3/547/ pdf - A Tug of War: Pseudorabies Virus and Host Antiviral Innate Immunity "

Lines 82-83: it is suggested to indicate the prevalence and incidence data of the disease in China.

Line 86: with regard to the human cases reported, it is suggested to provide valid references which highlights the real risk considering that the bibliographic evidence is limited to the categories shown. Although the risk exists it is suggested to provide a real one, the same is irrelevant and in any case limited to the categories exposed.

Line 93: indicate the origin and type of samples used for the identification and isolation of the two variants under study.

Line 99: indicate which type of vaccine (strain? Live vaccine? Attenuated vaccine?) And better clarify what is meant by "clinical pigs".

Line 108: it is suggested to make a brief introduction on the purposes of the experimentation in mice (evaluation of pathogenicity and evaluation of the immune response)

Lines 113-114: please indicate the strengths of the two inoculated PRV variants and the correct subdivision of the groups

Line 122: Please provide information about the euthanasia protocol used.

Lines 126-127: with reference to the acronym "SPF", this should be inserted in line 126 as it is mentioned first. Indicate the total number of pigs recruited.

Line 136: Please specify which clinical signs were monitored and how often.

Line 142: Please specify that the tonsils were taken from pigs only.

Line 148: Please specify which data were analyzed and the purpose for which the statistical methods indicated were chosen.

Lines 153-154: Please indicate the type of samples used for isolation, it is also suggested to move the entire period to the materials and methods section (see comment on line 93).

166-167: please move the entire period in materials and methods sections. 

Lines 169-178: the entire period denotes an interpretation of the results; therefore it is suggested to include it in the discussion section.

Line 191: the entire section is missing in the materials and methods section, please check.

Line 193: with regard to cell lines, indicate which types were used. Furthermore, as regards the PRV-TJ variant, used as a positive control, please specify the selection criterion for this variant. Furthermore, in the abstract section it is cited but not mentioned that it is a positive control, it is suggested to verify and clarify this aspect.

Line228: it is suggested to create a section in materials and methods, also the results of the statistical analysis are not indicated, please check.

Line 335: this conclusion is improper considering that an observational study has not been carried out, it is suggested to discuss the qualitative information, which the article provides and therefore: "isolation of two new variants and their characterization in terms of pathogenicity and efficacy of the immune response of vaccinated pigs ".

Lines 337-338: this statement, in an absolute sense, is inappropriate since it must be specified that the outcome of the infection is always fatal for non-natural hosts (species other than swine and wild boar). Although the literature reports human cases of encephalitis with serious consequences in humans, the correlation between death and PRV infection has not yet been demonstrated.

Reviewer 3 Report

Dear Editor,

The manuscript entitled: “Isolation and characterization of two pseudorabies virus and evaluation of their effects on host natural immune responses and pathogenicity” Zhou et al., describes the isolation and characterization of two (PRV-GC; PRV-JM) fatal PRV variants. The results showed that they belong to genotype 2.2. In addition, antisera from sows immunized with PRV classical vaccination evidenced much lower neutralization ability to PRV-GD and PRV-JM. However, the antisera from the pigs infected with PRV-JM had significantly higher neutralization ability to PRV-TJ, PRV-GD and PRV-JM. In vivo, PRV-GD and PRV-JM infection caused 100% death in mice and piglets and induced extensive tissue damage, cell death and inflammatory cytokines release. The results evidenced that pigs immunized with the classical PRV vaccine are incapable of providing sufficient protection against the PRV isolates, and there is a risk of continuous evolution and virulence enhancement.

I think this manuscript can be worth publishing if the following points are inserted:

General comments

1) Please, indicate if an Ethics Committee has authorized the experiments;

2) Please, in the manuscript, change the term “Suid herpesvirus 1” with “Suid alphaherpesvirus 1” in accordance with the last report of the International Committee on Taxonomy of Viruses (ICTV, 2020);

3) Please, insert Figure 1 A, Figure 1 D, Figure 2 A, Figure 5 C, Figure 6 D, Figure 7 A, B into the supplementary material for a better view of the pictures;

4) Please, in the Materials and Methods section, transfer the “Isolation and identification of PRV-GD and PRV-JM isolates” part gives in section 3.1;

5) Please, to identify PRVs, why have the fluorescent monoclonal antibodies not been used?

6) Please, describe the farms where the viruses were isolated and the type of sample used for viral isolation.

Specific comments

1) Introduction section (page 1, line 35): Please, the number 1 citation is not appropriate. I recommend including the International Taxonomy of Viruses (ICTV) as a citation source. (https://talk.ictvonline.org/ictv-reports/ictv_online_report/dsdna-viruses/w/herpesviridae/1609/subfamily-alphaherpesvirinae)

2) Experimental mice infection study (page 3, line 116): Please, insert the number of the citation Reed-Muench;

3) Proliferation characteristics of PRV-GD and PRV-JM in vitro (page 5, lines 194-195): Please, describe in extenso the cell used (PAMs, THP-1);

4) Immune responses induced by PRV strains (page 7, lines 229-231); Please transfer this paragraph in the materials and methods section;

5) Reference section (page 13): Please insert the 31st

Author Response

Review 2

The manuscript entitled: “Isolation and characterization of two pseudorabies virus and evaluation of their effects on host natural immune responses and pathogenicity” Zhou et al., describes the isolation and characterization of two (PRV-GD; PRV-JM) fatal PRV variants. The results showed that they belong to genotype 2.2. In addition, antisera from sows immunized with PRV classical vaccination evidenced much lower neutralization ability to PRV-GD and PRV-JM. However, the antisera from the pigs infected with PRV-JM had significantly higher neutralization ability to PRV-TJ, PRV-GD and PRV-JM. In vivo, PRV-GD and PRV-JM infection caused 100% death in mice and piglets and induced extensive tissue damage, cell death and inflammatory cytokines release. The results evidenced that pigs immunized with the classical PRV vaccine are incapable of providing sufficient protection against the PRV isolates, and there is a risk of continuous evolution and virulence enhancement.

I think this manuscript can be worth publishing if the following points are inserted:

 General comments

Q1. Please, indicate if an Ethics Committee has authorized the experiments;

Authors’ Response:

According to your suggestion, we added the Ethics Committee in the revised manuscript. Please check the changes in lines 417-420.

Q2. Please, in the manuscript, change the term “Suid herpesvirus 1” with “Suid alphaherpesvirus 1” in accordance with the last report of the International Committee on Taxonomy of Viruses (ICTV, 2020);

Authors’ Response:

Thanks for your professional comments. According to you suggestion, we changed the term “Suid herpesvirus 1” with “Suid alphaherpesvirus 1” in the revised manuscript (line 34).

Q3. Please, insert Figure 1 A, Figure 1 D, Figure 2 A, Figure 5 C, Figure 6 D, Figure 7 A, B into the supplementary material for a better view of the pictures;

Authors’ Response:

Thank you for your suggestion. After discussing with other authors, we all agree to keep the figures. For example, the figure 7 will not intact if the Figure 7A-B are removed.

Q4. Please, in the Materials and Methods section, transfer the “Isolation and identification of PRV-GD and PRV-JM isolates” part gives in section 3.1;

Authors’ Response:

According to your suggestion, the corresponding paragraph in section 3.1 had been transferred in the “materials and methods”. Please see the changes (Lines 97-101) in the revised manuscript.

Q5. Please, to identify PRVs, why have the fluorescent monoclonal antibodies not been used?

Authors’ Response:

It is a good suggestion. To identify PRV isolates, indirect immunofluorescence assay of glycoprotein B protein has been performed with anti-gB monoclonal antibody as the following Figure R1. We did not add it in the manuscript because of the limited space.

Figure R1. Indirect immunofluorescence assay of glycoprotein B (gB; red) proteins in PRV-TJ, PRV-GD and PRV-JM-challenged PK-15. The nucleus was stained with DAPI.

Q5. Please, describe the farms where the viruses were isolated and the type of sample used for viral isolation.

Authors’ Response:

We isolated the virus from the aborted piglets of PRV positive pig farms. We added the information in the revised manuscript. Please see the changes (line 99) in our revised manuscript.

Because we signed the confidentiality agreement with the farms, so we could not describe the farms where the viruses were isolated.

Specific comments

Q6. Introduction section (page 1, line 35): Please, the number 1 citation is not appropriate. I recommend including the International Taxonomy of Viruses (ICTV) as a citation source. (https://talk.ictvonline.org/ictv-reports/ictv_online_report/dsdna-viruses/w/herpesviridae/1609/subfamily-alphaherpesvirinae)

Authors’ Response:

Much appreciated for your professional comments, we added the website in the revised manuscript (Lines 36-37).

Q7. Experimental mice infection study (page 3, line 116): Please, insert the number of the citation Reed-Muench;

Authors’ Response:

Thanks for your suggestion. We added the citation in the revised manuscript (Line 134).

Q8. Proliferation characteristics of PRV-GD and PRV-JM in vitro (page 5, lines 194-195): Please, describe in extenso the cell used (PAMs, THP-1);

Authors’ Response:

Good suggestion and much appreciated. According to your suggestion, the cell lines used in each experiment were described in extenso in the revised manuscript (Lines 209-214).

Q9. Immune responses induced by PRV strains (page 7, lines 229-231); Please transfer this paragraph in the materials and methods section;

Authors’ Response:

According to your suggestion, the corresponding paragraph was transferred in the “materials and methods” (section 2.3, lines 117-124) in the revised manuscript.

Q10. Reference section (page 13): Please insert the 31st

Authors’ Response:

Sorry for our mistake. We removed the invalid citation and added the 31st reference in the revised manuscript.

Round 2

Reviewer 2 Report

The article entitled "Isolation and characterization of two pseudorabies viruses and evaluation of
their effects on natural immune responses and on host pathogenicity ", deals with a topic of great scientific interest, given the economic and potential zoonotic implications, due to the circulation of new variants of the PRV virus, towards which common vaccines could be not effective. After the revisions, the article is smoother, some limitations and shortcomings in the introductions, materials and methods and results sections have been overcome. The overall rating is good and sufficient for publication.
